# OpenReview forum: "Automatic Instance Selection with Genetic Updating for Few-shot LLM Jailbreak"
_ICLR.cc/2026/Conference — Submitted to ICLR 2026_

### Official Review · Reviewer_7z1H · 2025-10-29

**Soundness:** 4
**Presentation:** 4
**Contribution:** 3
**Rating:** 8
**Confidence:** 4

**Summary:**

The paper proposes a framework named ACCEPT for few-shot LLM jailbreaking. The method leverages textual gradients to automatically select the most effective instances and uses a genetic algorithm to enhance the harmful prompt through non-semantic perturbations. Experiments show that the proposed approach achieves superior results on both open-source and closed-source models.

**Strengths:**

1. The approach is highly innovative and logically sound; the concept of leveraging an LLM to generate "textual gradients" is particularly novel.
2. The experimental evaluation is thorough, and the accompanying analysis is detailed.
3. All figures and tables are aesthetically pleasing and easy to understand.

**Weaknesses:**

1. While the paper's primary focus is on offensive jailbreak techniques, the manuscript would be strengthened by briefly discussing the defensive implications of this work. For instance, adding a short elaboration in the conclusion on how these findings can inform the development of more robust defense mechanisms would be a valuable addition.
2. It would benefit from more specific details in the methodology sections. In particular, the descriptions of the TextGrad in Section 3.2 and the GA in Section 3.3 could be expanded with more concrete implementation details to improve clarity.

**Questions:**

1. In Section 2.1, the distinction between the two methods, "Adversarial Prompting and In-Context Learning," could be further detailed.
2. When the harmful request is combined with the selected instances for a jailbreak attack, how is it handled if the attack is not effective?
3. How are the emojis inserted into the harmful request, specifically the ones in Figure 7, selected? Inappropriate emojis could have a negative effect.
4. What does "fitness" mean in Figure 3?

---

> ### Author Response · Authors · 2025-11-17
> **Response to the comments**
>
> We greatly appreciate your review and will respond to each of your comments below. Additionally, we have uploaded the revised PDF version, and the corresponding modifications in the text have been marked in blue font.
>
> **Weakness 1: Elaboration of defense.**
>
> **Answer**：Thank you for your valuable suggestion. In response to your comment, we have added the following description in the conclusion to highlight the defensive implications of our work:
>
> > Importantly, our findings reveal the effectiveness of this hybrid optimization attack, which also provides insights for future defense strategies: defense systems need not only to recognize malicious semantic prompts in context but also to effectively filter or resist perturbations caused by non-semantic tokens.
>
> **Weakness 2: The descriptions of the TextGrad and the GA**
>
>
>
> **Answer:** Thank you for your valuable feedback. Regarding the details for TextGrad and GA, we have added the following conceptual descriptions (based on the original paper ) to the first paragraph of Section 3.2 and Section 3.3, respectively.
>
> > **For TextGrad:** Specifically, it first calculates the loss value of the current instance combination and uses a text-based gradient update mechanism to extract optimization suggestions that guide instance selection. These gradients are generated by a LLM to help identify the most favorable instances for attack effectiveness. **For GA:** Specifically, each perturbation strategy is encoded as a chromosome. GA iteratively generates new perturbation strategies through selection, crossover, and mutation operations, and evaluates the effectiveness of each strategy using a fitness function (i.e., the rejection response probability). In each generation, the most adaptive strategy is used to generate the next generation of perturbation schemes.
>
> **Question 1: In Section 2.1, the distinction between the two methods, "Adversarial Prompting and In-Context Learning," could be further detailed.**
>
> **Answer:** Thank you for the suggestion. We have added the following supplementary explanation in Section 2.1 to more clearly distinguish these two attack paradigms:
>
> > In short, the core distinction is: **Adversarial Prompting** focuses on manipulating the intrinsic properties of a single, standalone harmful prompt (e.g., adding obfuscating suffixes or designing deceptive scenarios) to directly circumvent safety alignments ; whereas **In-Context Learning Attacks** leverage the model's in-context learning ability, guiding the model to produce unsafe outputs by prepending the harmful request with a set of malicious question-answer instances.
>
> **Question 2: When the harmful request is combined with the selected instances for a jailbreak attack, how is it handled if the attack is not effective?**
>
> **Answer:** Thank you for your question. **In our framework, both the harmful request and the selection of instances are subject to optimization.** Therefore, if the attack is ineffective after combining the two, we first optimize the markers inserted into the harmful request using a Genetic Algorithm within an internal loop. If the attack is still ineffective after the internal loop, we further optimize the instances using TextGrad in an external loop. By optimizing from both perspectives, we ultimately improve the attack success rate.
>
> **Question 3: How are the emojis inserted into the harmful request, specifically the ones in Figure 7, selected? Inappropriate emojis could have a negative effect.**
>
> **Answer: (1) The insertion of emojis follows predefined rules.** Each rule is represented by a "chromosome" in the Genetic Algorithm (composed of a sequence of numbers), and the different numbers in the chromosome determine whether emojis are inserted, where they are placed in the harmful request, and how many times they appear. A detailed description of this process can be found in Section 3.3 under **Encoding of Perturbation Strategies** part;
>  **(2) Inappropriate emojis can indeed have a negative impact.** For example, an angry emoji might intensify the expression of the harmful request. Therefore, when selecting emojis, we avoid using malicious emojis and prefer neutral (e.g., books, gears) or benign (e.g., smiling faces) emojis, as shown in Figure 7.
>
> **Question 4: What does "fitness" mean in Figure 3?**
>
> **Answer**: The "fitness" in Figure 3 is a core metric used to **quantify the effectiveness of the perturbation strategy and the selected instance.** A higher fitness value indicates that the target model is more likely to respond to the harmful request. The specific calculation method is provided in Equation (5) in the main text.

---

> > ### Comment · Reviewer_7z1H · 2025-11-27
> >
> > Thanks for your responses. My concerns have been well addressed. I will maintain my score.

---

> > > ### Author Response · Authors · 2025-11-27
> > >
> > > Dear Reviewer,
> > >
> > > Thank you for your feedback and for confirming that our responses have addressed your concerns. We appreciate your time and valuable comments on our manuscript.

---

### Official Review · Reviewer_hs9X · 2025-10-29

**Soundness:** 3
**Presentation:** 3
**Contribution:** 3
**Rating:** 4
**Confidence:** 5

**Summary:**

This paper addresses the in-context learning-based few-shot jailbreak problem and proposes a unified framework named ACCEPT. The framework introduces two complementary components. For instance selection, ACCEPT employs a TextGrad mechanism that reformulates the discrete selection of few-shot examples into an optimization process guided by textual feedback from LLMs. For prompt design, it integrates a genetic algorithm to search for effective injection strategies, enhancing the stealthiness and adaptability of jailbreak prompts. Comprehensive experiments conducted on multiple open-source and closed-source language models, across various datasets, demonstrate that ACCEPT achieves SOTA attack performance, significantly surpassing existing jailbreak baselines.

**Strengths:**

1. ACCEPT tackles the few-shot jailbreak problem, which is both practically significant and highly relevant to the security of current LLM systems.
2. The use of a genetic algorithm for prompt injection provides a heuristic yet interpretable optimization process. This design allows researchers to better understand how perturbations influence jailbreak success, improving both transparency and controllability.
3. The TextGrad mechanism leverages LLM feedback to guide instance selection, transforming a discrete combinatorial problem into a continuous optimization-like process.
4. Extensive experiments across multiple open-source and closed-source models show consistent and substantial performance improvements, validating the effectiveness and robustness of the proposed framework.

**Weaknesses:**

1. Although the TextGrad mechanism achieves promising optimization results through LLM-based feedback, the paper lacks a solid theoretical justification. Its convergence properties and robustness are not formally analyzed or discussed.
2. The study primarily relies on the ASR-GPT metric, using an LLM to judge attack success. This single-metric setup may introduce bias from the evaluator model itself.
3. The proposed framework integrates multiple heuristic search processes (e.g., the genetic algorithm and iterative LLM optimization), which are likely to incur significant computational overhead. However, the paper provides no discussion or analysis of efficiency or resource consumption.

**Questions:**

Please refer to my comments on weaknesses.

---

> ### Author Response · Authors · 2025-11-17
> **Response to the comments**
>
> We greatly appreciate your review and will respond to each of your comments below. Additionally, we have uploaded the revised PDF version, and the corresponding modifications in the text have been marked in blue font.
>
> **Weakness 1: Although the TextGrad mechanism achieves promising optimization results through LLM-based feedback, the paper lacks a solid theoretical justification. Its convergence properties and robustness are not formally analyzed or discussed.**
>
> **Answer**: Thank you for this insightful point. The TextGrad mechanism is driven by LLMs, and its effectiveness is not based on strict mathematical convergence proofs, but rather on a core assumption: that the LLM, acting as the "gradient engine", possesses sufficient reasoning ability to evaluate a variable (referred to as "instances" in our paper), diagnose its issues, and provide meaningful improvement suggestions. **Therefore, the analysis of TextGrad's convergence and robustness ultimately depends on the LLM's capabilities.** We plan to further explore this area of research in future work. Although we currently lack formal theoretical proof, we have provided **empirical evidence** in the "Quantitative analysis of TextGrad" section of our main paper (as shown in Figure 3), which demonstrates that applying TextGrad does lead to steadily improving and gradually stabilizing effectiveness as the epochs increase.
>
> **Weakness 2: The study primarily relies on the ASR-GPT metric, using an LLM to judge attack success. This single-metric setup may introduce bias from the evaluator model itself.**
>
> **Answer**: Thank you for your valuable comment. To further validate the effectiveness of our method, we used the **Llama-Guard-3-8B** model to verify the output results from the AdvBench dataset. The experiment involved two target models and was tested under 2/8-shot conditions, with all other hyper-parameters matching those in the main experiments of the paper. The results, as shown in the table below, demonstrate that our method still achieves the optimal jailbreak success rate.
>
> | Traget Model | Qwen-2.5-7B (2-/8-shot) | **GPT-4o (2-/8-shot)** |
> | ------------ | ----------------------- | ---------------------- |
> | GCG          | 5.96% / 11.35%          | 8.08% / 13.65%         |
> | PANDAS       | 4.42% / 8.27%           | 7.50% / 15.96%         |
> | I-FSJ        | 3.65% / 7.31%           | 5.38% / 10.77%         |
> | ACCEPT       | **13.27% / 23.08%**         | **14.04% / 26.35%**        |
>
> **Weakness 3: The proposed framework integrates multiple heuristic search processes (e.g., the genetic algorithm and iterative LLM optimization), which are likely to incur significant computational overhead. However, the paper provides no discussion or analysis of efficiency or resource consumption.**
>
> **Answer:** Thank you for this valuable question. To quantify the efficiency of our method, we conducted a comparative experiment. We selected GCG and I-FSJ as baselines, as they also require iterative optimization, and tested on AdvBench50 against Qwen-2.5-7B (2-shot). The experimental results (shown in the table below) indicate that our method is **superior in terms of average attack time (Avg. Time (s))** compared to these two baselines, while also **maintaining a significant lead in ASR-GPT**.
>
> Nonetheless, we acknowledge that the overall time complexity of the ACCEPT framework (comprising $n$ TextGrad iterations and $t$ GA iterations, i.e., $O(n \times t)$) is still relatively high. Therefore, reducing the time complexity of the optimization process and improving overall efficiency will be an important direction for our future work. All the above analyses have been presented in Appendix C (Time Complexity Analysis) of the revised PDF.
>
> | Metrics       | GCG   | I-FSJ | ACCEPT    |
> | ------------- | ----- | ----- | --------- |
> | ASR-GPT(%)    | 16.00 | 20.00 | **46.00** |
> | Avg. Time (s) | 252   | 192   | **180**   |

---

### Official Review · Reviewer_nnRX · 2025-10-30

**Soundness:** 3
**Presentation:** 4
**Contribution:** 4
**Rating:** 8
**Confidence:** 4

**Summary:**

This paper introduces ACCEPT, an automated attack framework designed to address the challenge of instance selection in few-shot LLM jailbreak attacks. It achieves a superior jailbreak success rate through the synergistic cooperation of two components: a semantic-level module that selects optimal instances and a non-semantic-level module that enhances the attack prompt.

**Strengths:**

1. The overall framework is novel, enhancing both the selected instances and the malicious request from two synergistic perspectives: semantic and non-semantic.
2. The paper is well-written with a clear and coherent logical flow.
3. The experiments are extensive and sound.

**Weaknesses:**

1. The implementation of TextGrad relies on an LLM, so the quality of the LLM determines the final jailbreak success rate. It is necessary to elaborate on this part in the conclusion or appendix.
2. The paper should provide a clearer case study to illustrate how the "problem diagnosis" and "contribution assessment" parts of the textual gradient are generated and how they guide instance selection.

**Questions:**

1. In "Gradient-Guided Candidate Sampling", why select a candidate subset first instead of directly selecting from the entire sample pool C?
2. What does the single-point in the crossover operation mean?
3. In Algorithm 1, should the final output be E_{best}? I think that E_{T_{grad}} is not necessarily the best.
4. In the "Fitness Function," when calculating the probability for refusal terms, is it the sum of the refusal probabilities for each word, or is the refusal probability calculated for the entire sentence?

---

> ### Author Response · Authors · 2025-11-17
> **Response to the comments**
>
> We greatly appreciate your review and will respond to each of your comments below. Additionally, we have uploaded the revised PDF version, and the corresponding modifications in the text have been marked in blue font.
>
> **Weakness 1：The implementation of TextGrad relies on an LLM, so the quality of the LLM determines the final jailbreak success rate. It is necessary to elaborate on this part in the conclusion or appendix.**
>
> **Answer:** Thank you for your comment. We have added the following content in the conclusion:
>
> > Notably, the implementation of TextGrad relies on the performance of the LLM. Therefore, the quality of the selected LLM directly affects the final jailbreak success rate. A higher-quality LLM can provide more accurate text gradient feedback, which improves the optimization of instance selection and increases the success rate of the attack.
>
> **Weakness 2: Case Study is needed.**
>
> **Answer**: Thank you for your comment. To address this, we have provided a clear case study in the **Qualitative analysis of TextGrad** part in Section 4. Additionally, the entire process is depicted in detail in Figure 4 of the main text.
>
> **Question 1: In "Gradient-Guided Candidate Sampling", why select a candidate subset first instead of directly selecting from the entire sample pool C?**
>
> **Answer**: The sample pool $C$ contains thousands of examples, and if all these examples were processed by the LLM at once, **it would significantly increase computational resource consumption and time**. Therefore, in our implementation, we first select a candidate subset that is highly relevant to the current malicious requests. This approach not only reduces resource consumption but also improves the jailbreak success rate.
>
> **Question 2: What does the single-point in the crossover operation mean?**
>
> **Answer**: In the crossover mechanism of Genetic Algorithms, the single-point operation refers to selecting two "parent" chromosomes (which encode perturbation strategies), randomly choosing a common position (i.e., the "single point") in their gene sequences, and then swapping all the gene segments after that point. This generates two new offspring individuals.
>
> **Question 3: In Algorithm 1, should the final output be E_{best}? I think that E_{T_{grad}} is not necessarily the best.**
>
> **Answer:** Thank you very much for your correction. We have replaced $E\_{T\_{grad}}$ with $E_{best}$ in the revised version.
>
> **Question 4: In the "Fitness Function," when calculating the probability for refusal terms, is it the sum of the refusal probabilities for each word, or is the refusal probability calculated for the entire sentence?**
>
> **Answer:** The probability of the refusal terms refers to the refusal probability of the **entire sentence** (or phrase), not the sum of the refusal probabilities of individual words. This is because individual words from a refusal phrase might appear in a harmful response even in a successful jailbreak scenario, which would lead to a situation where the jailbreak was successful, but the final refusal probability is also high. To prevent this situation, we calculate the probability of the entire sentence.

---

> > ### Comment · Reviewer_nnRX · 2025-11-26
> >
> > Thanks for your response. The response has addressed all my concerns, and I will maintain my rating.

---

> > > ### Author Response · Authors · 2025-11-26
> > >
> > > Dear Reviewer,
> > >
> > > Thank you for your feedback and for confirming that our responses have addressed your concerns. We appreciate your time and valuable comments on our manuscript.

---

### Official Review · Reviewer_Jt2U · 2025-10-31

**Soundness:** 2
**Presentation:** 3
**Contribution:** 2
**Rating:** 6
**Confidence:** 3

**Summary:**

This paper devises a few-shot jailbreak method called ACCEPT. The goal is to provide the target LLM with some instances together with a harmful question and leverage the in-context learning to make it output a harmful response. The method starts with a set of candidate instances and utilizes an auxiliary LLM to provide "textual gradient" to narrow down the candidate pool and select new instances. To reduce the refusal rate, it also includes a genetic algorithm to inject special non-semantic tokens. The instance selection and the genetic algorithm run alternately. Experiments use two datasets, AdvBench and HarmBench, and six models against five baselines. Results show the proposed method can achieve best ASRs and is more robust against defenses. Ablation studies show that both the instance selection and the genetic algorithm are necessary.

**Strengths:**

1. It studies a critical issue of few-short jailbreaks.

2. The idea to combine the textual gradient and the genetic algorithm is interesting.

3. The experimental results show its effectiveness and robustness.

**Weaknesses:**

1. It's unclear how the fitness score is computed. Although the paper lists some refuse phrases in the appendix, it's not clear how they are used to compute the probability. For example, does it generate a set of samples and compute the ratio of the matched strings, or use some token distribution? If the token distribution is used, how can we attack the closed-source model or service that doesn't return the distribution?

2. It's unclear how the quality of the candidate pool will affect the performance. Because TextGrad only selects the instances from the pool without creating any new instances, and only one candidate pool was used. For example, if the algorithm starts with some random pool instead of the malicious ManyHarm, it may not succeed.

3. It's unclear what the time cost is or the queries needed to conduct this attack. Similarly, it's suggested to compare the efficiency with baselines.

4. Regarding the robustness, since the injected non-semantic tokens include some special tokens, paraphrasing or input filtering may easily remove the adversarial effect. Also, it's unclear what the baseline is in Figure 2.

**Questions:**

1. How is the fitness score computed?

2. Will a different candidate pool affect the performance significantly?

3. What is the time cost and the query budget?

---

> ### Author Response · Authors · 2025-11-17
> **Response to the comments**
>
> We greatly appreciate your review and will respond to each of your comments below. Additionally, we have uploaded the revised PDF version, and the corresponding modifications in the text have been marked in blue font.
>
> **Weakness 1: It's unclear how the fitness score is computed.**
>
> **Answer:** Thank you for your question. **For the fitness score:** In our framework, the fitness score is obtained by calculating the probability of target token sequences (i.e., the "refuse phrases" in the appendix) appearing during the generation process. We use the **token probability distribution** (rather than string matching ratio) to compute the joint probability of these refusal phrases appearing in the model's output. **For the closed-source model** (such as GPT-4o used in our paper), your point is very accurate. These models indeed do not return the full vocabulary distribution. In practice, we use API requests (e.g., setting `logprobs=True` and `top_logprobs=10`) to obtain the most probable (e.g., Top-10) tokens and their probabilities at each generation step. Therefore, our probability calculation is based on this returned **partial distribution** (as an approximation).
>
> **Weakness 2: It's unclear how the quality of the candidate pool will affect the performance.**
>
> **Answer:** Thank you for raising this critical point. **We acknowledge that the quality of the candidate pool is an important factor and has a certain correlation with performance.** Our currently used ManyHarm candidate pool ($C$) contains nearly 2,400 data entries across 12 high-risk topics, providing a sufficiently large and diverse search space. To further verify the performance impact you mentioned, we selected the HarmBench dataset as the candidate pool (as a random candidate pool, representing a reference group), then used AdvBench50 as the harmful requests, and tested on three models and in 2/8-shot settings (evaluation metric is ASR-GPT). **The results below show that when HarmBench was used as the candidate pool, there was a slight decrease in performance.** However, compared to the baselines in Table 1 of the main text, the performance of ACCEPT-HarmBench is still the best.
>
> |**Target Model**|**Qwen-2.5-7B**(2/8-shot) | **Llama-3.1-8B** (2/8-shot)|GPT-4o (2/8-shot)|
> |----------------| --------------------------|--------------------------- |----------------- |
> |ACCEPT-ManyHarm|31.54%/41.15%|7.69%/12.12%|41.35%/78.65%|
> |ACCEPT-HarmBench|28.17%/37.63%|7.21%/10.72%|35.20%/69.54%|
>
> **Weakness 3: Time cost analysis.**
>
> **Answer:** Thank you for this valuable question. To quantify the efficiency of our method, we conducted a comparative experiment. We selected GCG and I-FSJ as baselines, as they also require iterative optimization, and tested on AdvBench50 against Qwen-2.5-7B (2-shot). The experimental results (shown in the table below) indicate that our method is **superior in terms of average attack time (Avg. Time (s))** compared to these two baselines, while also **maintaining a significant lead in ASR-GPT**.
>
> Nonetheless, we acknowledge that the overall time complexity of the ACCEPT framework (comprising $n$ TextGrad iterations and $t$ GA iterations, i.e., $O(n \times t)$) is still relatively high. Therefore, reducing the time complexity of the optimization process and improving overall efficiency will be an important direction for our future work. All the above analyses have been presented in Appendix C (Time Complexity Analysis) of the revised PDF.
> |Metrics|GCG|I-FSJ|ACCEPT|
> |-------------|-----|-----|---------|
> |ASR-GPT(%)|16.00|20.00|**46.00**|
> |Avg. Time (s)|252|192|**180**|
>
> **Weakness 4: Robustness**
>
> **Answer: (1) Regarding robustness, we acknowledge your point that operations like filtering will eliminate the special characters we introduced**. This is reflected in our **+PPL** defense test (Figure 2 in the main text). When the shot is 4, our method is indeed affected, leading to a final result lower than the Baseline. However, in most cases (such as when shot=2 and 8), our method is still effective. Furthermore, our method actually introduces both optimized instance selection and special character insertion simultaneously. Even if the non-semantic markers are filtered, the semantic instances optimized by TextGrad remain efficient. It can be seen in the ablation study (Table 2 in the main text) that when only the instance selection strategy is used (TextGrad-Only), our method still outperforms the Baseline. **(2) The Baseline in Figure 2 refers to the method of randomly selecting *n* instances as a prefix and not applying any processing to the harmful request.**
>
> **Question 1, 2, and 3.**
>
> We greatly appreciate you for raising Questions 1, 2, and 3. The answers can be found in response to Weakness 1, Weakness 2, and Weakness 3, respectively.

---

> ### Comment · Reviewer_Jt2U · 2025-11-26
>
> I appreciate the authors' reply. I keep my score.

---

> > ### Author Response · Authors · 2025-11-27
> >
> > Dear Reviewer,
> >
> > Thank you for your feedback and for confirming that our responses have addressed your concerns. We appreciate your time and valuable comments on our manuscript.

---

### Meta-Review · Area_Chair_z5Td · 2026-01-07

**Summary:**

This paper studies jailbreak to LLMs, with a focus on few-shot settings. The idea is to design an Automatic Instance Selection method to select few shot instances to improve the effectiveness of attacks. The paper also performs experiments to validate the effectiveness of the proposed method.

**Reviewer Concerns:**

In general, this paper studies the safety issues of LLMs, which is an important and popular topic. The idea of selecting few shot instances to improve the effectiveness of attacks also makes sense. While three out of four reviewers give positive scores, I have some concerns after reading the paper, reviews, and responses. First, the technical novelty of the method is unclear. The paper combines TextGrad (a popular method for prompt optimization) and a genetic-based algorithm. However, the genetic-based algorithm is widely used to perform jailbreak attacks to LLMs. The proposed method is not compared with these methods (such as AutoDan, TAP). The paper may consider comparing with state-of-the-art jailbreak methods. Based on Table 1, the ASR of the proposed method is very low, e.g., around 10% on Llama-3.1-8B. Other papers (e.g., JAILBREAKING LEADING SAFETY-ALIGNED LLMS WITH SIMPLE ADAPTIVE ATTACKS , ICLR’25) already show very high ASRs on many LLMs. Second, in Table 1, only six LLMs are evaluated, with one closed-source LLM (GPT-4o). Many other popular open-source and closed-source LLMs are not evaluated (e.g., Gemini, Deepseek, Llama-2). In general, the size of LLMs is mostly around 8B. The current results are far from comprehensive to demonstrate the effectiveness of the proposed method. There are also other concerns from reviewers, such as efficiency. In the response, the paper only compares the efficiency on a single setting with two baselines (GCG, I-FSJ). The paper may perform a more comprehensive evaluation (at least compare with all baselines) and discuss the efficiency of the proposed method. Given the above concerns, the current version of the paper may not be ready for publication.

**Reviewer Scores:**

The reviewers may not change their scores.

---

### Decision · Program_Chairs · 2026-01-26

Reject